# Bifunctionalized Gold Nanoparticles for the Colorimetric Detection of the Drug γ-Hydroxybutyric Acid (GHB) in Beverages

**Silvia Rodríguez-Nuévalos** [1], **Ana M. Costero** [1,2,*] , **Salvador Gil** [1,2] , **Margarita Parra** [1,2] **and Pablo Gaviña** [1,2,*]

1    Instituto Interuniversitario de Investigación de Reconocimiento Molecular y Desarrollo Tecnológico (IDM), Universidad Politècnica de València, Universitat de València, Doctor Moliner 50, 46100 Burjassot, Spain; Silvia.rodriguez@uv.es (S.R.-N.); salvador.gil@uv.es (S.G.); Margarita.Parra@uv.es (M.P.)
2    CIBER de Bioingeniería, Biomateriales y Nanomedicina (CIBER-BBN), 28029 Madrid, Spain
*    Correspondence: Ana.Costero@uv.es (A.M.C.); pablo.gavina@uv.es (P.G.); Tel.: +34-963543740 (P.G.)

**Abstract:** The increase in the number of drug-facilitated sexual assault (DFSA) cases in recent years has become a major concern. Consequently, there is a need to develop methods for the real-time detection of these substances. We report herein a colorimetric chemosensor for the real-time in situ detection of the "date rape" drug GHB. The sensor is based on gold nanoparticles functionalized with both a 2-aminonaphthoxazole and phenanthroline derivative. Its ability to act as "naked-eye" colorimetric sensor for the detection of the drug in soft drinks and alcoholic beverages was studied. The detection process is based on the double recognition of both the hydroxyl and the carboxylate groups present in GHB, which triggers the aggregation of the AuNPs, with the resulting change in the color of the solution.

**Keywords:** γ-hydroxybutyric acid (GHB); gold nanoparticles; colorimetric sensing; 2-aminonaphthoxazol; phenanthroline

## 1. Introduction

According to the European Monitoring Centre for Drugs and Drug Addiction (EM-CDDA), the number of cases of "drug-facilitated sexual assaults" (DFSA) has increased in recent years. Six European countries have carried out population surveys in order to know the extent of this problem, and the results have shown that around 20% of women had experienced some sexual assault during their adulthood. However, the exact number of cases remains unknown due to the lack of efficient monitoring systems to detect this situation. The most commonly used drugs in DFSA are central-nervous-system depressants such as alcohol, benzodiazepines, ketamine or γ-hydroxybutyric acid (GHB) [1]. Apart from alcohol, GHB is one of the most used drugs for two main reasons. First, it is an odorless and colorless compound that exhibits a slight salty taste in water solution, so it can be easily delivered to the victims without them realizing it [2]. Second, its detection after consumption is elusive, since it is fast metabolized in the body ($C_{max}$ is reached 20–40 min after its ingestion). It is eliminated from plasma with a half-life of 30–50 min and only small amounts of this substance can be recovered from victim's urine (1–5% of the dose). Thus, the drug detection time is very short (3–10 h) [3]. For the above reasons, it is highly desirable and necessary to develop efficient detection systems to prevent this type of aggressions, as well as to detect them in a reasonable short time after they have taken place.

GHB detection methods are being widely studied, but most are based on chromatographic [4,5] and spectroscopic methods [6], which, demonstrating high sensitivity, usually require complicated samples treatments. The use of optical sensors is usually a simple and much faster alternative [7]. These probes can detect the analyte in real time, and special

skills are not necessary to use them. Despite this, only a few chromo-fluorogenic chemosensors for GHB recognition have been described, with very diverse approaches [8–12]. The latest system published by Xing and coworkers [13] is based on a specific enzymatic reaction of GHB that generates NADH, which facilitates the reduction of Au (III) to give rise to AuNPs, producing a color change.

Gold nanoparticles (AuNPs) have received a great amount of attention over the past years as scaffolds to prepare chemosensors to sense different analytes. Their usefulness is based on their optoelectronic properties, with one of the most remarkable properties being their surface plasmon resonance (SPR) [14]. The SPR absorption band can be modified by different factors, such as AuNPs shape, size or aggregation state. Thus, under an appropriate stimulus, well-dispersed AuNPs can aggregate, which results in a bathochromic shift of the SPR absorption band with the corresponding change in the color of the solution from red (dispersed) to blue (aggregated). This color change usually can be observed by the naked eye. This fact, in addition to their biocompatibility, as well as their easy surface functionalization with many different types of organic molecules, has converted AuNPs in a common and useful material to develop rapid, economic and selective colorimetric sensors [15].

Taking all this into account and the expertise of our group in the development of chemosensors based on functionalized AuNPs [16,17], here we report a selective and efficient method to detect GHB in beverages. The sensing protocol is based on the use of AuNPs doubly functionalized with two different ligands to recognize the functional groups present in GHB: a carboxylate and an alcohol group. The recognition units chosen were a 2-aminonaphthoxazole moiety that is capable of interacting with the carboxylate groups [18] and a phenanthroline group to interact with the hydroxyl group of GHB [19,20]. We expected that GHB would trigger the aggregation of the gold nanoparticles through a double recognition process. The recognition paradigm is shown in Scheme 1.

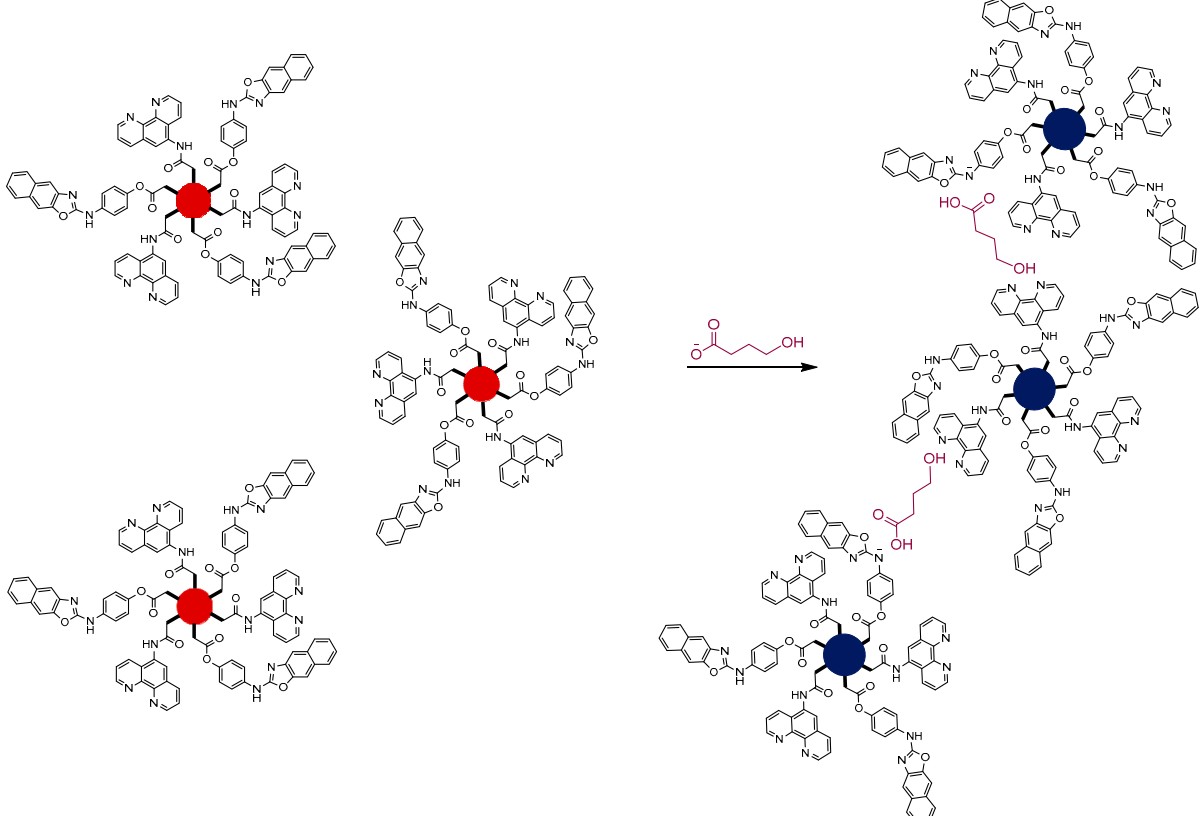

**Scheme 1.** Proposed paradigm for GHB recognition with sensor GNP1.

## 2. Materials and Methods

The reagents employed in the synthesis were acquired from Sigma-Aldrich and used without further purification. $^1$H NMR and $^{13}$C NMR spectra were registered in Bruker Avance 300 or 500 MHz spectrophotometers, all of them referenced to solvent peak, DMSO($d_6$). UV–Vis spectra were registered in a Shimadzu UV-2600 spectrophotometer, using a cuvette with 1 cm of path length. All measurements were carried out at room temperature. Images of transmission electron microscopy were taken with JEOL-1010 transmission electron microscopy, operating at 100 kV. Z potential and DLS values were measured in a Malvern Zetasizer ZS, Malvern United Kingdom, for 3 times, in 10–25 cycles. Mass Spectrometry Spectra were carried out with a TripleTOFTM 5600 LC/MS/MS System, with 2 gas sources (both to 35 psi), 450 °C and ion gas voltage of 5500 V. Origin 2020 was the program to plot titrations.

### 2.1. Synthesis of the Ligands

#### 2.1.1. Synthesis of 3

In a 50 mL 3-neck round bottom-flask, 201 mg of 3-amino-2-naphthol (1.26 mmol) was dissolved in 9 mL of pyridine under an argon atmosphere. Then 180 μL of 1-isothiocyanato-4-methoxybenzene was added, and the mixture was kept stirring overnight, at room temperature. After that, the solvent was removed, and the brown oil obtained was dissolved in 30 mL of AcOEt. Then the organic phase was washed with 15 mL of 1% aqueous HCl, 15 mL of saturated aqueous $NaHCO_3$ and 15 mL of saturated aqueous NaCl, successively. The organic phase was dried over anhydrous $MgSO_4$, filtered and the solvent removed to vacuum. The reaction crude was purified by chromatography column, using silica gel as stationary phase and Hexane:AcOEt from 7:3 to 5:5 as eluent. Then 350 mg of compound **3** was obtained as a brown solid in 96% yield. $^1$H NMR (300 MHz, DMSO) δ 10.48 (s, 1H), 10.06 (s, 1H), 9.16 (s, 1H), 8.84 (s, 1H), 7.74–7.60 (m, 2H), 7.39 (d, *J* = 9.0 Hz, 2H), 7.35–7.22 (m, 2H), 7.19 (s, 1H), 6.95 (d, *J* = 9.0 Hz, 2H), 3.76 (s, 3H).

#### 2.1.2. Synthesis of 4

In a 25 mL 2-neck round-bottom flask, 181 mg of **3** (0.56 mmol) was dissolved in 5 mL of THF. Then 127 μL of $H_2O_2$ 30% (1.12 mmol) and 3 mg of TBAI (0.008 mmol) were added. The stirring was kept overnight at room temperature. Then the solvent was removed, and the brown oil obtained was dissolved in 15 mL of AcOEt. The organic phase was washed with 15 mL of deionized water, dried over anhydrous $MgSO_4$ and filtered. After removing solvent, 160 mg of compound 4 was obtained as a brown solid (99% yield). $^1$H NMR (500 MHz, DMSO) δ 10.67 (s, 1H), 7.97–7.91 (m, 2H), 7.90 (s, 1H), 7.81 (s, 1H), 7.71 (d, *J* = 9.0 Hz, 1H), 7.45–7.37 (m, 2H), 7.00 (d, *J* = 9.0 Hz, 2H), 3.76 (s, 3H). $^{13}$C NMR (126 MHz, DMSO) δ 159.61, 155.02, 147.15, 143.07, 131.47, 131.31, 129.45, 127.63, 127.38, 124.37, 123.89, 119.69, 114.28, 111.89, 104.35, 55.27.

#### 2.1.3. Synthesis of 5

In a 100 mL round-bottom flask, 59 mg of compound **4** (0.20 mmol) was solved in 3 mL of DCM and cooled at 0 °C. Then 820 μL of $BBr_3$ (0.82 mmol) was added dropwise. The reaction temperature was risen to room temperature, and the stirring was kept overnight. After that, 15 mL of deionized water was added, and the stirring was kept for 15 min. DCM was removed, and compound **5** was extracted with AcOEt (3 × 15 mL). The organic phase was washed with 15 mL of saturated aqueous NaCl, dried over $MgSO_4$, filtered and the solvent was removed to vacuum. Compound **5** was obtained as a brown solid (60 mg, 99% gross yield). $^1$H NMR (300 MHz, DMSO) δ 10.52 (s, 1H), 9.24 (s, 1H), 7.97–7.88 (m, 2H), 7.88 (s, 1H), 7.77 (s, 1H), 7.57 (d, *J* = 9.0 Hz, 2H), 7.45–7.37 (m, 2H), 6.81 (d, *J* = 9.0 Hz, 2H). $^{13}$C NMR (126 MHz, DMSO) δ 159.77, 153.14, 147.22, 143.31, 131.32, 129.95, 129.38, 127.60, 127.34, 124.31, 123.80, 119.93, 115.47, 111.69, 104.23.

### 2.1.4. Synthesis of **L1**

In a 50 mL 2-neck round-bottom flask, under argon atmosphere, 46 mg of (±)-α-lipoic acid (0.22 mmol), 10 mg of DMAP (0.081 mmol) and 40 μL of EDC were solved in 2 mL of dry THF. The stirring was kept for 1 h at room temperature. Then 60 mg of **5** was solved in 4 mL of dry THF and added to the previous mixture. The reaction was stirred overnight at room temperature. After that, the solvent was removed, and the resulting brown oil was solved in 15 mL of AcOEt. Then the organic phase was washed with 10 mL of acid aqueous solution (pH 4), 10 mL of saturated aqueous $NaHCO_3$ and 10 mL of saturated aqueous NaCl. The organic phase was dried over anhydrous $MgSO_4$, filtered and the solvent removed to vacuum. The reaction crude was purified by chromatography column, using silica gel as stationary phase and Hexane:AcOEt 7:3 as eluent. Then 51 mg of **L1** was obtained as a brown powder (51% yield). $^1$H NMR (500 MHz, DMSO) δ 10.93 (s, 1H), 8.00–7.89 (m, 3H), 7.86 (s, 1H), 7.84 (d, *J* = 9.0 Hz, 2H), 7.48–7.39 (m, 2H), 7.17 (d, *J* = 9.0 Hz, 1H), 3.67–3.60 (m, 1H), 3.22–3.16 (m, 1H), 3.15–3.08 (m, 2H), 2.58 (t, *J* = 7.3 Hz, 2H), 2.45–2.38 (m, 1H), 1.94–1.84 (m, 1H), 1.76–1.55 (m, 4H), 1.51–1.41 (m, 2H). $^{13}$C NMR (126 MHz, DMSO) δ 171.91, 159.31, 149.62, 147.03, 145.53, 145.50, 142.84, 136.13, 135.99, 131.31, 129.61, 127.67, 127.49, 124.44, 124.07, 123.91, 122.35, 122.32, 118.87, 112.36, 104.55, 56.06, 38.14, 34.05, 33.29, 28.08, 24.14. HRMS: m/z calculated for $C_{25}H_{25}N_2O_3S_2$ (M + H): 465.1307; found: 465.1301 $[M + H]^+$.

### 2.1.5. Synthesis of **L2**

In a 50 mL 2-neck round-bottom flask, 100 mg of 11-mercaptoundecanoic acid (0.46 mmol) was solved in 10 mL of dry DCM. Then 40 μL of $SOCl_2$ (0.47 mmol) was added, and the mixture was heating to reflux for 6 h. After that, the stirring was kept overnight to room temperature. Next, the solvent was removed to vacuum, and 158 mg of $K_2CO_3$ (1.15 mmol) and 2 mL of acetone were added to the crude of reaction. After 15 min, 89 mg of 1,10-phenanthrolin-5-amine (0.46 mmol) and 2 mL of acetone were added to the previous mixture, and the stirring was kept for 4 h. Then the solid was filtered at vacuum, re-suspended in 10 mL of water deionized and 10 mL of AcOEt for 15 min and filtered at vacuum. Finally, 60 mg of **L2** was obtained as a light brown powder (30% yield). $^1$H NMR (500 MHz, DMSO) δ 10.24 (s, 1H), 9.10 (dd, *J* = 4.3, 1.6 Hz, 1H), 9.01 (dd, *J* = 4.3, 1.7 Hz, 1H), 8.65 (dd, *J* = 8.4, 1.7 Hz, 1H), 8.44 (dd, *J* = 8.1, 1.8 Hz, 1H), 8.17 (s, 1H), 7.81 (dd, *J* = 8.4, 4.3 Hz, 1H), 7.74 (dd, *J* = 8.1, 4.3 Hz, 1H), 2.86 (t, *J* = 7.2 Hz, 0.7H), 2.83–2.77 (m, 0.3H), 2.66 (t, *J* = 7.2 Hz, 1H), 2.54 (t, *J* = 7.3 Hz, 2H), 1.77–1.63 (m, 3H), 1.58 (p, *J* = 7.2 Hz, 1H), 1.46–1.12 (m, 12H). $^{13}$C NMR (126 MHz, DMSO) δ 172.55, 149.89, 149.32, 145.66, 143.54, 135.96, 131.92, 128.13, 124.69, 123.67, 122.88, 119.89, 38.15, 37.89, 35.96, 34.05, 28.88, 28.86, 28.81, 28.75, 28.56, 28.50, 28.21, 27.70, 27.66, 25.26. HRMS: m/z calculated for $C_{46}H_{57}N_6O_2S_2$ (M + H): 789.3984; found: 789.3967 $[M + H]^+$.

## 2.2. *Synthesis of the Functionalized Gold Nanoparticles (AuNPs)*

### 2.2.1. Synthesis of Citrate-Capped Gold Nanoparticles (Citrate-GNPs)

Prior to use, the whole material was washed with aqua regia and dried at 120 °C, in an oven, for 24 h. In a 250 mL 3-neck round-bottom flask, 39.4 mg of $HAuCl_4$ (0.1 mmol) was dissolved in 100 mL of Milli-Q water and boiled. Then a solution of 114.1 mg of trisodium citrate dihydrate (0.38 mmol) in 10 mL of Milli-Q water was quickly added, and the resulting mixture was boiled and vigorously stirred for 30 min. After that, the mixture was cooled to room temperature, and the AuNPs were preserved until their use in the fridge, at 4 °C.

### 2.2.2. Synthesis of GNP1

In a 25 mL round-bottom flask, 3900 μL of Milli-Q water was mixed with 21 μL of aqueous NaOH 0.5 M for 1 min. Then 2100 μL of the previously prepared AuNPs was added. After that, **L1** (15 μL, 0.5 mM) and **L2** (15 μL, 0.32 mM) were added simultaneously, and the stirring was kept for 1 h. Next, the mixture was diluted with 6 mL of Milli-Q

water and centrifuged for 10 min at 10,500 rpm. Then the supernatant was discarded and replaced by 6 mL of Milli-Q water.

### 2.2.3. Exchange of Buffer

First, 3 mL of GNP1 was diluted with 3 mL of Milli-Q water, divided in six Eppendorf (1 mL of mixture per Eppendorf) and centrifuged at 10,500 rpm for 10 min. The supernatants were discarded and replaced by 1 mL of buffer (1 mM Phosphate pH 6.5, 7.5 and 8.5) or deionized water. The solutions were kept in the fridge at 4 °C for 5 days.

### *2.3. UV–Vis Measurements*

### 2.3.1. GHB Titration

Prior to measure, each solution of NaGHB was prepared from 1 M aqueous NaGHB solution. Concentrations ranged from 0 to 100 mM. Then 400 µL of each solution was mixed with 400 µL of GNP1 in a 1 mL cuvette, and, after an incubation period of 2 min, the UV–Vis spectra were registered.

### 2.3.2. AcONa/EtOH vs. GHB

In a 1 mL cuvette, 400 µL of GNP1 was mixed with 400 µL of 70 mM aqueous AcONa/EtOH or 70 mM aqueous NaGHB, and, after an incubation period of 2 min, the UV–Vis spectra were registered.

### 2.3.3. Interferent Measurements

In a 1 mL cuvette, 400 µL of GNP1 was mixed with 400 µL each interferent (0.3% $w/v$ citric acid and 0.01% $w/v$ sodium ascorbate), and, after an incubation period of 2 min, the UV–Vis spectra were registered.

### *2.4. Real Samples*

Previously, beverage samples were adulterate with NaGHB, in a 70 mM concentration. Then 100 µL of sample was mixed with 100 µL of GNP1, at room temperature. The same process was followed for the samples without GHB. The changes were observed immediately.

## 3. Results and Discussion

### *3.1. AuNPs Preparation*

The structure and syntheses of both recognition units (**L1** and **L2**) anchored onto the gold surface are depicted in Schemes 2 and 3, respectively. Ligand **L1** consists of a lipoic acid derivative incorporating a 2-aminonaphthoxazole moiety. First, thiourea **3** was prepared by reaction between 3-amino-2-naphthol and isothiocyanate-4-methoxy-benzene in pyridine [21]. Then **3** was treated with $H_2O_2$ in presence of a catalytic amount of TBAI to obtain the naphthoxazole derivative **4** [22]. Demethylation of **4** with $BBr_3$ in DCM gave rise to phenol **5**. Finally, esterification between **5** and lipoic acid in the presence of EDC and a catalytic amount of DMAP led to ligand **L1** (Scheme 2), which incorporates a disulfide group to anchor the ligand to the gold surface [23].

**L2**, incorporating two 5-amido-1,10-phenanthrolines connected through a disulfide bridge, was synthesized as depicted in Scheme 3. 11-mercaptoundecanoic acid was converted in its correspondent acid chloride derivative by using $SOCl_2$ in DCM. Then the **L2** was obtained by reaction with 5-amino-1,10-phenanthroline. Under the reaction conditions, the disulfide bond was spontaneously formed by oxidation of the thiols.

On the other hand, AuNPs were prepared following the method of Turkevich–Frens, which provides AuNPs whose diameters range from 15 to 150 nm [24]. Firstly, citrate-stabilized gold nanoparticles (citrate-GNPs) were prepared by reducing tetrachloroauric acid with trisodium citrate in boiling water. Secondly, the functionalization of the citrate-GNPs with ligands **L1** and **L2** to obtain GNP1 was optimized by varying the quantity of

gold nanoparticles, the amount of both ligands, and the reaction and the centrifugation time (see Supplementary Materials).

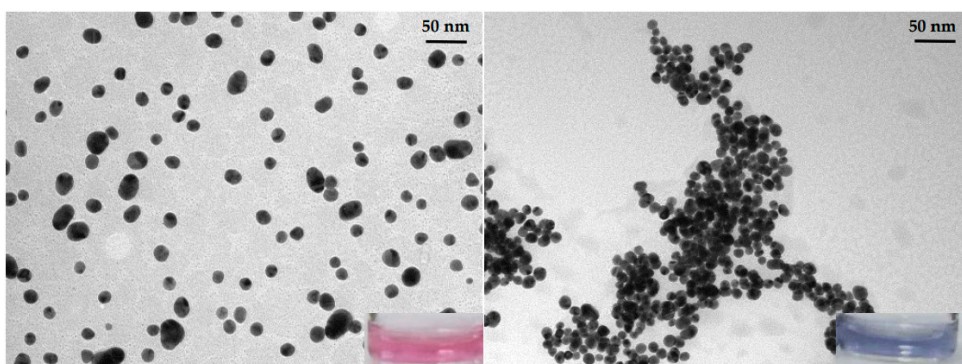

**Scheme 2.** Synthetic pathway designed to obtain **L1**.

**Scheme 3.** Synthetic pathway designed to obtain **L2**.

The diameter of GNP1, determined from TEM images (Figure 1), was 18 nm. Data from DLS (Dynamic Light Scattering) and zeta-potential corresponding to citrate–GNP (the gold nanoparticles citrate-capped diluted to 35%) and GNP1 are summarized in Table 1 (GNP1 distribution size is shown in the Supplementary Materials).

**Figure 1.** Changes induced in the aggregation of GNP1 in presence of GHB.

**Table 1.** Size and zeta-potential measured for citrate–GNP and GNP1 in deionized water.

| Material | Size (nm) TEM Images | Size (nm) DLS | Z Potential (mV) |
| --- | --- | --- | --- |
| Citrate–GNP | —— | $24.2 \pm 1.6$ | $-29.2 \pm 0.9$ |
| GNP1 | $18 \pm 2$ | $25.2 \pm 0.3$ | $-37 \pm 2$ |

GNP1 has a hydrodynamic diameter slightly higher than citrate–GNP, which can be related to the ligand interchange. The small change observed can be related to the low functionalization of GNP1. On the other hand, Z-potential of GNP1 is also higher than this

value for citrate–GNP. These data indicated a higher stabilization of GNP1 when compared with the nanoparticles without functionalization.

The concentration of GNP1 turned out to be $(5.7 \pm 0.4) \cdot 10^{-10}$ M. This was calculated from UV–Vis spectra measurements, considering $6.01 \times 10^8$ M$^{-1} \cdot$cm$^{-1}$ as an estimated molar extinction coefficient [25].

The stability of GNP1 in different media was evaluated. Several buffer solutions (1 mM phosphate, pH 6.5, 7.5 and 8.5) and deionized water were used, and the suspensions were kept in the fridge for 5 days. After this time, only the suspensions in deionized water remained dispersed (Supplementary Materials Figure S8). In consequence, this was the medium used in the sensing experiments.

### 3.2. Sensing Experiments

In order to verify the detection ability of sensor GNP1 towards GHB, a preliminary UV–Vis study was developed. GNP1 exhibits an absorption band centered at 525 nm in the UV–Vis spectrum, which is in accordance with the SPR band of dispersed gold nanoparticles of ca 20 nm diameter [15]. An addition of an excess of GHB, after 2 min of incubation time, promoted the aggregation of the nanoparticles with a bathochromic shift of the plasmon band to 643 nm and a clear change in the color of the solution from red to blue (Figure 2). The position of the maximum in the UV–Vis spectrum, in addition to the color change, strongly suggests that aggregation of GNP1 was taking place. This was further supported by TEM images that clearly showed the aggregation (Figure 1).

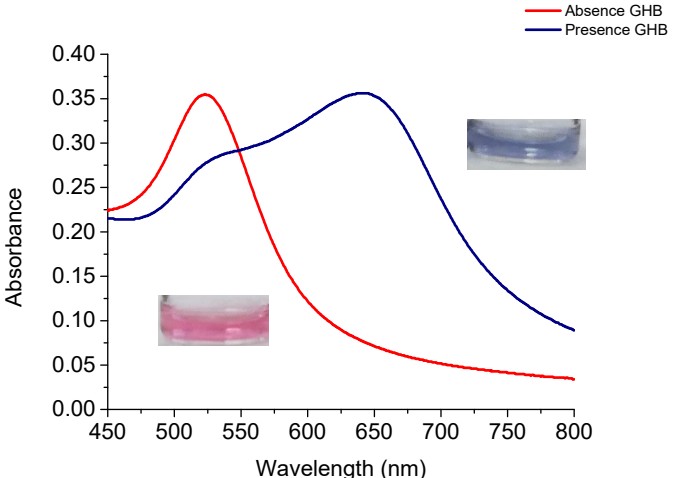

**Figure 2.** GNP1 UV–Vis spectra variation in absence and presence of 100 mM of GHB, using deionized water as solvent.

UV–Vis titrations of GNP1 using increasing amounts of GHB were made in deionized water (see Figure 3). A decrease of the band at 525 nm and an enhancement of the band at 643 nm was observed as the concentration of GHB increased. These changes induced a gradual change in the color of the suspension, which could be observed by the naked eye.

The probe showed a linear response from 16 to 28 mM. From these data and using the expression LoD = $3 \cdot S_b / m$ (where $S_b$ is the standard deviation of the blank, and m is the slope), a value of LoD = 1.12 mM was determined. At this point, it is worth noting that the average amount of GHB necessary to facilitate anesthesia, leading to severe confusional episodes and coma, is 50 mg/kg [26]. It means that a person who weighs 62 kg (the average body mass) has to intake about 3100 mg of GHB to promote a coma state. That implies an approximate concentration of 90 mM, a higher value than the LoD found.

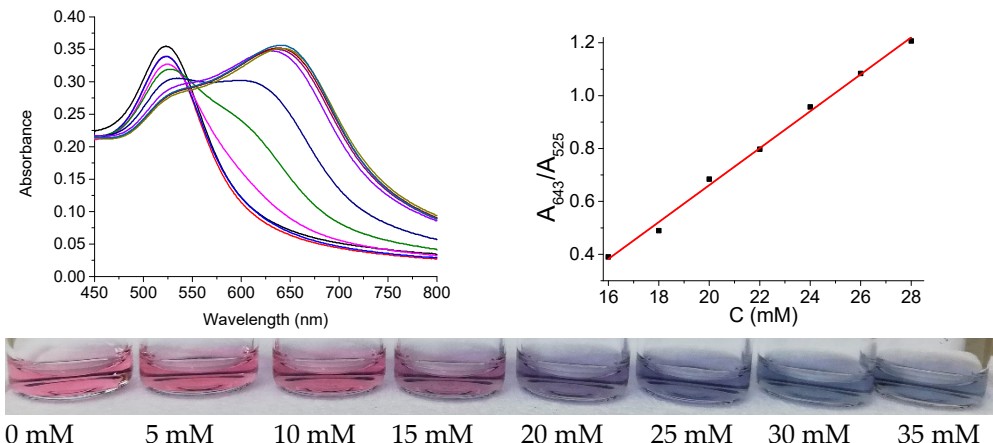

**Figure 3.** (**Top**) Titration of GHB ranging from 0 to 35 mM with GNP1 (**left**) and its linear regression (**right**, y = (0.0698 ± 0.0019)·x − 0.74 ± 0.04, $R^2$ = 0.996), using deionized water as solvent. (**Bottom**) Colorimetric changes observed during the titration.

To demonstrated that both the hydroxyl and the carboxylate groups of GHB are simultaneously involved in the aggregation process, the responses of the sensor in the presence of GHB (35 mM) and a mixture of AcONa and EtOH (35 mM each) were evaluated (see Supplementary Materials Figure S9). A slight change was observed in the UV–Vis spectrum of GNP1 when the mixture AcONa/EtOH was added, but it was not as remarkable as the response induced by GHB. This result reinforces the interaction of both functional groups with the receptors units designed.

As the main aim of this work was to test GNP1 in beverages, the response in presence of citric acid and sodium ascorbate was evaluated, as these compounds are present in soft drinks, where their concentrations are usually around 0.3% *w/v* and 0.01% *w/v*, respectively [27]. Citric acid, due to its acidity, caused a destabilizing interaction, because negative charges which surrounded GNP1 were partially neutralized, promoting its rapid aggregation. On the other hand, sodium ascorbate had the opposite effect, acting as a stabilizing agent. Thus, when both compounds are together (in the described concentrations), ascorbic acid partially counteracts the negative effect of citric acid, allowing for the detection of GHB (see Supplementary Materials Figure S10).

The last step was to test sensor GNP1 in real samples. For this, detection of GHB in some beverages, such as alcoholic drinks and soft drinks (alone and mixed with some distilled drink), was studied. Initially, the samples were spiked with GHB (70 mM final concentration) and mixed with GNP1 at room temperature (for further details, see "Section 2"). In all the tested cases, the changes were immediately observed by the naked eye. The results were satisfactory, as can be seen in Figure 4, and it was possible to distinguish between clean samples and samples adulterated with GHB (35 mM final concentration). Apart from orange soda and whisky, other samples were tested and compiled in Supplementary Materials Figure S11.

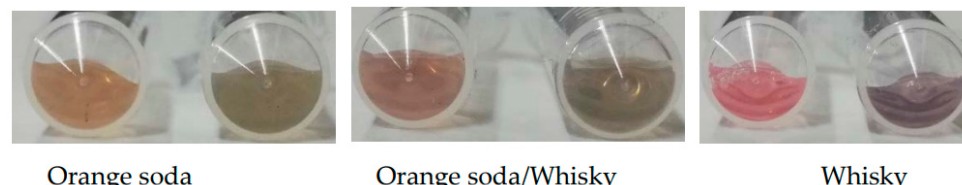

**Figure 4.** Color changes in real samples. Please notice that, in all cases, the vials are grouped in pairs, and the first vial contains the clean sample; meanwhile, the second one contains the spiked sample.

The visual observation of the drug in beverages clearly suggests that the system could be easily modified to be used onsite. In this sense, a kit for personal safety to be used in

recreational environments could be envisaged. This kit would be formed by an Eppendorf containing the AuNPs suspension and a simple instructions sheet (an example is included in Supplementary Materials Figure S12). The addition of a drop of the beverage to the container, followed by the color observation, would permit the user to know if the beverage has been contaminated with GHB.

## 4. Conclusions

In conclusion, we report herein a colorimetric sensor (GNP1) for the quick and easy detection of GHB by the naked eye in both aqueous solutions and real samples. Sensor GNP1 consists of an aqueous dispersion of gold nanoparticles functionalized with two different ligands, a 2-aminonaphthoxazole derivative that is capable of interacting with carboxylate groups and a phenanthroline terminated ligand that can bind to hydroxyl groups. These ligands can recognize both functional groups present in GHB, giving rise to an aggregation process, which in turns results in a clear color change of the solution from red to blue and a bathochromic shift of the SPR band in the UV–Vis spectrum. The LoD here reported (1.12 mM) is considerable smaller than the quantity necessary of GHB to induce a severe confusion (90 mM). The system is suitable for the preparation of kits for personal safety to be used in recreational environments.

**Supplementary Materials:** The following are available online at https://www.mdpi.com/article/10.3390/chemosensors9070160/s1. Figure S1. $^1$H NMR spectrum of **L1**. Figure S2. $^{13}$C NMR spectrum of **L1**. Figure S3. $^1$H NMR spectrum of **L2**. Figure S4. $^{13}$C NMR spectrum of **L2**. Figure S5. Mass Spectrometry Spectrum of **L1**. Figure S6. Mass Spectrometry Spectrum of **L2**. Figure S7. GNP1 DLS size distribution. Figure S8. GNP1 stability in different buffers. Figure S9. UV-visible changes of sensor GNP1 with 35 mM GHB and a 35 mM mixture of AcONa/EtOH using water deionised as solvent. Figure S10. Absorbance spectra of GNP1 in the presence of citric acid, sodium ascorbate, in the real concentration present in beverages, and NaGHB (35 mM). Figure S11. Colour changes in real sample. Figure S12. Instructions sheet for an on-site safety kit based on GHB detection.

**Author Contributions:** Conceptualization, A.M.C.; methodology, S.R.-N.; characterization, S.G.; sources, M.P.; writing—review and editing, S.R.-N., A.M.C. and M.P.; supervision, P.G. All authors have read and agreed to the published version of the manuscript.

**Funding:** This research was funded by Spanish Government, MICINN funds (RTI2018- 100910-B-C42) and Ministry of Health, Consumer Affairs and Social Welfare Project 2020I040 PNSD 2020 and the Generalitat Valenciana (PROMETEO 2018/024).

**Institutional Review Board Statement:** Not applicable.

**Informed Consent Statement:** Not applicable.

**Data Availability Statement:** Not applicable.

**Acknowledgments:** S.R.-N. is grateful to the Spanish Government for a fellowship. SCSIE (Universidad de Valencia) is gratefully acknowledged for all the equipment used. NMR was registered at the U26 facility of ICTS "NAMBIOSIS" at the Universitat of València which is gratefully acknowledged.

**Conflicts of Interest:** The authors declare no conflict of interest regarding the publication of this paper.

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
