# Peer review of "Bifunctionalized Gold Nanoparticles for the Colorimetric Detection of the Drug γ-Hydroxybutyric Acid (GHB) in Beverages"

_chemosensors, doi:10.3390/chemosensors9070160_

Round 1

Reviewer 1 Report

This manuscript reports a bifunctionalized gold nanoparticles for the colorimetric detection of g-hydroxybutyric acid (GHB) in beverages by triggers the aggregation of the AuNPs. The author claimed that the sensor can monitor the target in the beverage in situ in real time. The key to this sensor is 2-aminonaphthoxazole and phenanthroline derivative functionalized gold nanoparticles, which is through the double recognition of both the hydroxyl and the carboxylate groups present in GHB. Its ability to act as “naked-eye” colorimetric detect to GHB in soft drinks and alcoholic beverages. However, major modification is requested in order to meet the requirement of Chemosensors journal as follows:

  1. The author claimed that the sensor can monitor the target in the beverage in situ in real time. How is this situation achieved? Please give the actual research process.
  2. The particle size distribution diagram of Au NPs should be given for further analysis.
  3. The recognition mechanism proposed by the author should be further experimentally proved.
  4. In Figure 1, TEM image of Au NPs and aggregated Au NPs can’t prove the cause of the aggregation caused by the target. In the aggregated gold nanoparticles, the particle size does not change significantly and still presents single particle shape. Is this caused by different sample preparation procedures? The author should provide further experimental proof. In addition, the resolution of these images is too low and the scale is not obvious, and should be further improved.
  5. The pH value, temperature and buffer solution may be important influencing factors for the sensor, and the author should provide relevant experimental optimization procedures.
  6. The selectivity, anti-interference ability, reproducibility, and stability of the sensor should be proved by experiments.
  7. The detection result of the sensor in the real sample should be presented.
  8. There are multiple data graphs in the Figure in the manuscript that should be numbered.

Author Response

We would like to thank reviewer 1 for his/her positive comments. In relation to his/her suggestions we have included several modifications in the revised version.

  1. The referee says “The author claimed that the sensor can monitor the target in the beverage in situ in real time. How is this situation achieved? Please give the actual research process. The experimental process used to carry out detection in real sample already was briefly explained in the manuscript. Anyway, for more clarity a most extended description has been included in material and methods.

  1. The referee indicates “The particle size distribution diagram of Au NPs should be given for further analysis”. The distribution diagram has been included in the supporting information (Figure S7).

  1. The referee comments “The recognition mechanism proposed by the author should be further experimentally proved”. The detection mechanism is based on the aggregation induced by the interaction of GHB with both functional groups present in the probe. The interaction of GHB with benzoxazole derivatives like this used in the present system has been widely demonstrated in our previous work published in Chem Comm (ref. 18) (before revison it was ref. 7). On the other hand, phenanthroline derivatives has been used frequently to recognize hydroxyl groups. To reinforce this behavior, in addition to ref. 19 (before revision ref. 8), reference 20 has been now included in the manuscript.

  1. The referee says “In Figure 1, TEM image of Au NPs and aggregated Au NPs can’t prove the cause of the aggregation caused by the target. In the aggregated gold nanoparticles, the particle size does not change significantly and still presents single particle shape. Is this caused by different sample preparation procedures? The author should provide further experimental proof. In addition, the resolution of these images is too low and the scale is not obvious, and should be further improved”. In relation to these comments, Figure 1 has been changed to show the scale clearly and increase quality. We agree with the referee that TEM experiments can’t prove the cause of the aggregation but only that this aggregation has taken place after the addition of GHB. As the effect is smaller when a mixture of ethanol and acetic acid are used in the text it is reasonable to suggest that the aggregation involved both functional groups simultaneously. These results suggest that the changes observed are not related to the preparation conditions.

  1. The referee indicates “The pH value, temperature and buffer solution may be important influencing factors for the sensor, and the author should provide relevant experimental optimization procedures”. Studies at different pH values have been done and the results have been included in the supporting information (Figure S8). The higher stability of the probe was observed in deionized water and as the stability of the system is important and the sensing results were similar in all the media, the experiments in real samples were carried out in water.

  1. The referee suggests “The selectivity, anti-interference ability, reproducibility, and stability of the sensor should be proved by experiments”. As it was indicated in point 5 the stability of the probe was evaluated. The final purpose of the sensor is detecting GHB in beverages. For this reason, the interferents studied were the compounds that are present in drinks such as citrate and ascorbic acid. A general study of other interferents seems not to be of interest for detecting the drug in beverages. Finally, all the experiments in real samples were repeated several times with the same results.

  1. The referee says, “The detection result of the sensor in the real sample should be presented”. The results with different soft and mixed beverages have been showed in the manuscript (Figure 4) and in the supporting information (Figure S11).

  1. The referee indicates “There are multiple data graphs in the Figure in the manuscript that should be numbered”. All the figures included in the manuscript have been revised and improved as much as possible.

Reviewer 2 Report

Authors have reported a facile and novel method for the detection of one of the commonly used drugs of abuse (GHB). I found their work very organized, clear sequence of ideas and highly interesting topic that is worthy of publication in your journal.

I have only one simple note for this work, which is to add previous detection methods for this drug especially by using metallic nanoparticles such as the work by Hu et. al. 2020 Chembiochem. DOI: 10.1002/cbic.202000157

Author Response

We would like to thank reviewer 2 for his/her for his/her positive comments. In relation to his/her suggestion the indicated reference has been included in the manuscript (Reference 13).

Reviewer 3 Report

Although the idea on which the work presented by the authors is based is not innovative, the application makes it extremely interesting, with a possible impact on society. The work is well written and the analysis clear. I only suggest 2 small changes in order to improve the manuscript.

  1. In the introduction, the authors say "The SPR absorption band can be modified by different factors such as AuNPs shape, size or aggregation state .." I suggest to add a comment on how even a change in the surrounding medium can greatly modify the position of the SPR band too.
  2. The linear regression reported in the inset of Figure 3 is totally unreadable. Since the figure allows it, I suggest that the authors make this inset as large as the absorbance graph .. they might indicate them as top left (Absorbance plot) and top right the linear regression.
  3. The bibliography not provide sufficient background, I suggest to improved it.

Author Response

We would like to thank reviewer 3 for his/her for his/her positive comments. In relation to his/her suggestions we included our responses

  1. The referee suggests “In the introduction, the authors say "The SPR absorption band can be modified by different factors such as AuNPs shape, size or aggregation state". I suggest to add a comment on how even a change in the surrounding medium can greatly modify the position of the SPR band too”. A new reference (14) according to this suggestion has been included in the test.

  1. The referee says “The linear regression reported in the inset of Figure 3 is totally unreadable. Since the figure allows it, I suggest that the authors make this inset as large as the absorbance graph they might indicate them as top left (Absorbance plot) and top right the linear regression”. Figure 1 has been modified according to the suggestions made by the referee.

  1. The referee indicates “The bibliography does not provide sufficient background, I suggest to improved it”. Several new references (4 to 14 and 20) have been included in the revised version.

Reviewer 4 Report

The manuscript “ Bifunctionalized gold nanoparticles for the colorimetric detection of the drug r-hydroxybutyric acid (GHB) in beverages” uses the principle of surface plasmon resonance shift by gold nanoparticles aggregation which cause a color change to detect GHB. The principle applied here is not novel, but this work is beneficial to society.

I don’t have any questions about the surface plasmon resonance principle applied here in this manuscript. Only a few places need to correct for the language.

Author Response

We would like to thank reviewer 4 for his/her positive comments. According to his/her indication the manuscript has been revised to correct possible language mistakes.

Round 2

Reviewer 1 Report

The authors have addressed well the comments and made all necessary changes. I recommend to accept this manuscript for publication.

Author Response

Thank you very much for your recomendation